# The representation of facial emotion expands from sensory to prefrontal cortex with development

Xiaoxu Fan[1], Abhishek Tripathi[2], Kelly Bijanki[1]*

[1]Baylor College of Medicine, Houston, United States; [2]Rice University, Houston, United States

## eLife Assessment

This study examines an **important** question regarding the developmental trajectory of neural mechanisms supporting facial expression processing. Leveraging a rare intracranial EEG (iEEG) dataset including both children and adults, the authors reported that facial expression recognition mainly engaged the posterior superior temporal cortex (pSTC) among children, while both pSTC and the prefrontal cortex were engaged among adults. In terms of strength of evidence, the **solid** methods, data and analyses broadly support the claims with minor weaknesses.

*For correspondence:
bijanki@bcm.edu

Competing interest: The authors declare that no competing interests exist.

**Abstract** Facial expression recognition develops rapidly during infancy and improves from childhood to adulthood. As a critical component of social communication, this skill enables individuals to interpret others' emotions and intentions. However, the brain mechanisms driving the development of this skill remain largely unclear due to the difficulty of obtaining data with both high spatial and temporal resolution from young children. By analyzing intracranial EEG data collected from childhood (5–10 years old) and post-childhood groups (13–55 years old), we find differential involvement of high-level brain area in processing facial expression information. For the post-childhood group, both the posterior superior temporal cortex (pSTC) and the dorsolateral prefrontal cortex (DLPFC) encode facial emotion features from a high-dimensional space. However, in children, the facial expression information is only significantly represented in the pSTC, not in the DLPFC. Furthermore, the encoding of complex emotions in pSTC is shown to increase with age. Taken together, young children rely more on low-level sensory areas than on the prefrontal cortex for facial emotion processing, suggesting that the prefrontal cortex matures with development to enable a full understanding of facial emotions, especially complex emotions that require social and life experience to comprehend.

## Introduction

Understanding others' emotional states through their facial expressions is an important aspect of effective social interactions throughout the lifespan. Behavioral data suggest that facial emotion processing emerges very early in life (*Barrera and Maurer, 1981*; *Walker-Andrews, 1997*), as infants just months old can distinguish happy and sad faces from surprised faces (*Caron et al., 1982*; *Nelson and Dolgin, 1985*; *Nelson et al., 1979*). However, children's emotion recognition is substantially less accurate than adults, and this ability prominently improves across childhood and adolescence (*Johnston et al., 2011*; *Kolb et al., 1992*; *Lawrence et al., 2015*; *Rodger et al., 2015*; *Romani-Sponchiado et al., 2022*; *Thomas et al., 2007*). Although extensive research in cognitive and affective neuroscience has

**eLife digest** Facial expressions are a primary source of social information, allowing humans to infer others' emotions and intentions. The ability to discriminate basic facial expressions emerges early in life, and infants can distinguish sad from happy faces. The ability to read facial expressions and to recognize emotions continues to improve from childhood through adolescence.

Most developmental evidence comes from behavioral measures or non-invasive imaging methods, with limited insight into the timing and distribution of neurons involved in these processes. This makes it difficult to characterize how neural representations of facial emotions change with age.

A method known as intracranial EEG (iEEG) provides direct, high-spatial and temporal resolution measurements of neural activity. It enables precise comparisons of how different brain regions encode facial emotion information across developmental stages, and how perceptual and cognitive systems mature to support social understanding. However, iEEG is rarely available in both children and adults due to its invasive nature.

Fan, Tripathi and Bijanki wanted to better understand how the brain learns to recognize emotions from facial emotions as children grow older by analyzing an existing, open-access iEEG data set. Specifically, they asked whether young children rely mainly on brain areas that process what a face looks like, while older individuals also use brain areas involved in interpretation and understanding.

Fan et al. found clear age-related differences in how the brain processes facial emotions. In young children, information about facial expressions is mainly processed in brain areas that analyze facial features. In older individuals, additional brain regions involved in interpretation and higher-level understanding also become engaged. The researchers also found that the brain's ability to distinguish complex emotions increases gradually with age. Together, these findings suggest that as children grow and gain social experience, the brain increasingly relies on higher-level systems to interpret the emotional meaning of faces, not just their visual features.

Understanding how recognizing emotions becomes more accurate and complex with age is important for clinicians, educators, and families concerned with children's social and emotional development – especially for conditions in which emotion understanding develops atypically. Before these findings can be applied in practice, future studies will need to confirm them in larger and more diverse groups and use non-invasive brain measures suitable for everyday settings. Translating these insights into tools for assessment or intervention will also require close collaboration between researchers, clinicians and educators.

assessed developmental changes using behavioral and non-invasive neuroimaging approaches, our understanding of brain development related to facial expression perception remains limited.

One influential perspective on the development of face recognition is that it depends on the maturation of face-selective brain regions, including the fusiform face area (FFA), occipital face area (OFA), and posterior superior temporal sulcus (pSTS) (*Duchaine and Yovel, 2015*). Supporting this view, *Gomez et al., 2017* found evidence for microstructural proliferation in the fusiform gyrus during childhood, suggesting that improvements in face recognition are a product of an interplay between structural and functional changes in the cortex. Additionally, monkeys raised without exposure to faces fail to develop normal face-selective patches, suggesting that face experience is necessary for the development of the face-processing network (*Arcaro et al., 2017*). It is likely that the gradual maturation of pSTS and FFA, two early sensory areas involved in the processing of facial expressions (*Bernstein and Yovel, 2015*; *Engell and Haxby, 2007*), contributes to the improved facial expression recognition over development. Yet, few studies have investigated the development of neural representation of emotional facial expressions in FFA and pSTS from early childhood to adulthood in humans.

Besides the visual processing of facial configurations, understanding the emotional meaning of faces requires the awareness and interpretation of the emotional state of the other person, which is significantly shaped by life experience (*Pereira et al., 2019*; *Yurgelun-Todd and Killgore, 2006*). Thus, some researchers have proposed that the maturation of emotional information processing is related to the progressive increase in functional activity in the prefrontal cortex (*Kolb et al., 1992*; *Tessitore et al., 2005*; *Williams et al., 2006*). With development, greater engagement of the prefrontal cortex may facilitate top-down modulation of activity in more primitive subcortical and limbic regions, such

as the amygdala (*Hariri et al., 2003*; *Hariri et al., 2000*; *Yurgelun-Todd, 2007*). However, despite these theoretical advances, the functional changes in the prefrontal cortex during the perceptual processing of emotional facial expressions over development remain largely unknown.

Here, we analyze intracranial EEG (iEEG) data collected from childhood (5–10 years old) and post-childhood groups (13–55 years old) while participants were watching a short audiovisual film. In our results, children's dorsolateral prefrontal cortex (DLPFC) shows minimal involvement in processing facial expression, unlike the post-childhood group. In contrast, for both children and post-childhood individuals, facial expression information is encoded in the pSTC, a brain region that contributes to the perceptual processing of facial expressions (*Bernstein and Yovel, 2015*; *Duchaine and Yovel, 2015*; *Fan et al., 2020*; *Flack et al., 2015*). Furthermore, the encoding of complex emotions in the pSTC increases with age. These iEEG results imply that social and emotional experiences shape the prefrontal cortex's involvement in processing the emotional meaning of faces throughout development, probably through top-down modulation of early sensory areas.

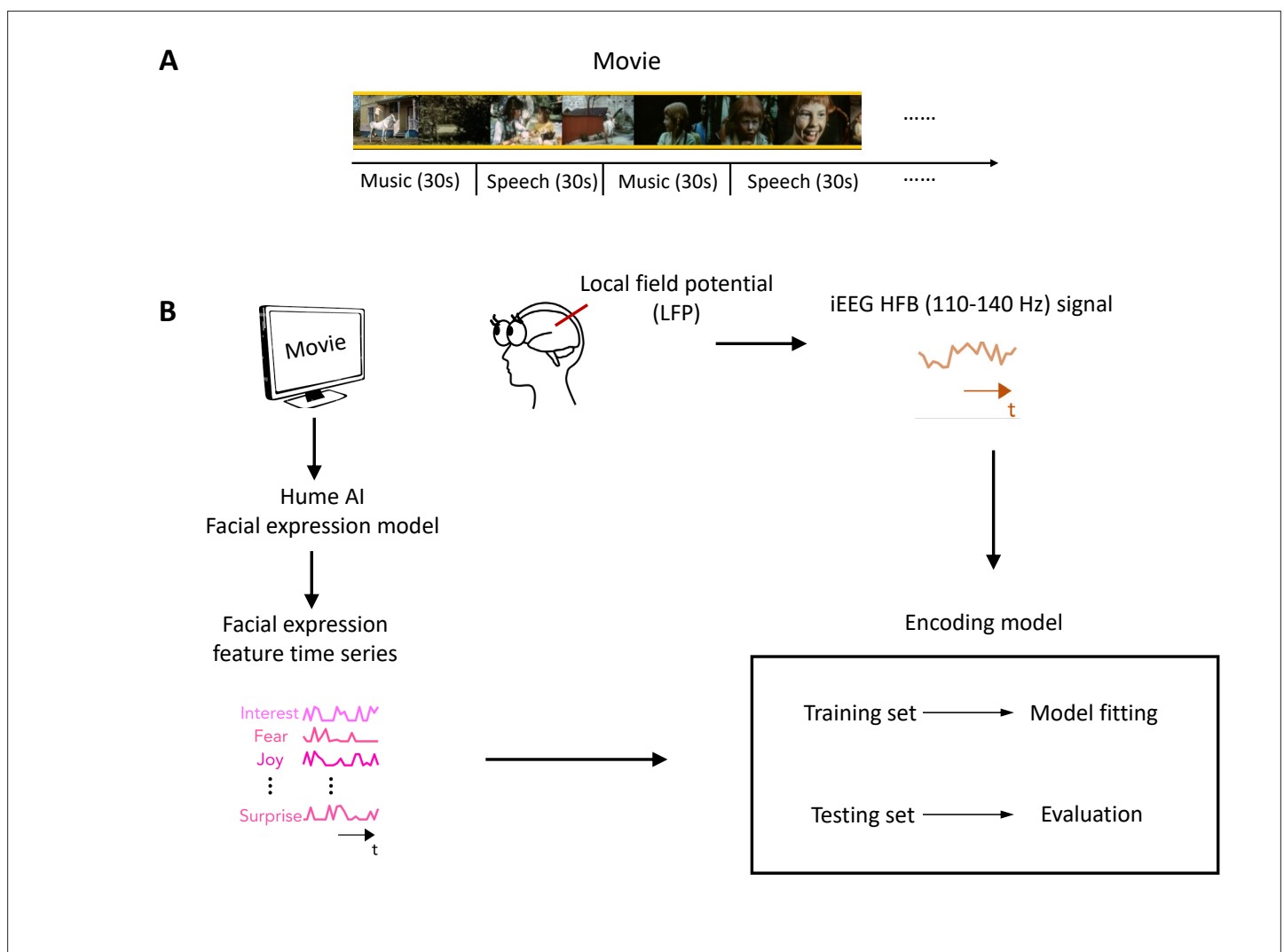

**Figure 1.** Task design and analysis methods. (**A**) Movie structure. A 6.5 min short film was created by editing fragments from *Pippi on the Run* into a coherent narrative. The movie consisted of 13 interleaved blocks of videos accompanied by speech or music. (**B**) Data analysis schematic. Standard analysis pipeline for extracting emotion features from the movie and constructing an encoding model to predict intracranial EEG (iEEG) responses while participants watch the short film.

## Results

### Using AI and encoding models to study the neural representation of facial expression

In this study, we analyzed iEEG data collected from a large group of epilepsy patients while they watched a short audiovisual film at the University Medical Center Utrecht (*Berezutskaya et al., 2022*). The movie consisted of 13 interleaved blocks of videos accompanied by speech or music, 30 s each (*Figure 1A*). To characterize the neural representation of facial expression in the prefrontal cortex and low-level sensory areas across development, we analyzed iEEG data from 11 children (5–10 years old) and 31 post-childhood individuals (13–55 years old) who have electrode coverage in DLPFC, pSTC, or both. First, Hume AI facial expression models were used to continuously extract facial emotion features from the movie (*Figure 1B*). Then, we tested how well encoding models constructed from the 48 facial emotion features (e.g. fear, joy) predict cortical high-frequency band (HFB) activity (110–140 Hz) induced by the presented movie (*Figure 1B*). We focused on HFB activity because it is widely considered to reflect the responses of local neuronal populations near the recording electrode (*Parvizi and Kastner, 2018*). The model performance was quantified as the correlation between the predicted and actual HFB activities, which is also called prediction accuracy.

### Differential representation of facial expression in children's DLPFC

Using the analysis approach described above, we examined how facial emotion information is represented by DLPFC (*Figure 2A*) while watching videos accompanied by speech (i.e. speech condition) in childhood and post-childhood groups. The prediction accuracy of the encoding model was significantly greater than zero in the post-childhood group (*Figure 2B*, gray bar, $p=0.012$, $t_{12}=2.954$, one-sample t-test), suggesting that the neural responses in DLPFC were dynamically modulated by the facial emotion features from the movie. However, facial emotion features were not encoded in children's DLPFC (*Figure 2B*, blue bar, $p=0.73$, $t_7=0.355$, one-sample t-test). The prediction accuracy in children's DLPFC was significantly lower than in the post-childhood group ($p=0.036$, unpaired t-test). Similar results were observed in the music condition. The prediction accuracy of the encoding model was significantly greater than zero in the post-childhood group (*Figure 2C*, $p=0.012$, $t_{12}=2.975$, one-sample t-test), but not in the childhood group (*Figure 2C*, $p=0.19$, $t_7=1.469$, one-sample t-test). Together, these findings show that the DLPFC dynamically encodes facial expression information in post-childhood individuals but not in young children.

To further understand the functional development of children's DLPFC, we compared the effect of human voice on the representation of facial expression in DLPFC between the two groups. The effect of human voice was quantified as a difference in prediction accuracy between the speech and music conditions. Our results showed that human voice influences facial expression representation in the DLPFC differently across development (*Figure 2D*, $p=0.018$, $t_{19}=2.588$, unpaired t-test). The presence of human voice enhances facial expression representation in the DLPFC of post-childhood individuals but impairs it in children.

Taken together, there are significant developmental changes in DLPFC involvement in facial expression perception.

### The neural representation of facial expression in children's pSTC

After identifying developmental differences in the involvement of high-level brain areas in processing facial expression, we next examined the neural representation of facial expression in children's early sensory areas. As an area in the core face network, pSTS has been associated with early stages of facial expression processing stream (*Bernstein and Yovel, 2015*; *Duchaine and Yovel, 2015*; *Fan et al., 2020*; *Flack et al., 2015*). Although previous studies suggested that the development of facial recognition depends on the maturation of face-selective brain regions (*Arcaro et al., 2017*; *Gomez et al., 2017*), it is still unclear how facial expression information is encoded in children's pSTS. Here, we examined the performance of the facial expression encoding model in a rare sample of children with electrode coverage in pSTC (*Figure 3A*). Our results showed that the encoding model significantly predicts the HFB neural signals in children's pSTC under the speech condition (*Figure 3B*, $p<0.002$, two-tailed permutation test, see **Method** for details). Moreover, the prediction accuracy is significantly reduced when human voice is absent from the video (*Figure 3B*, $p=0.0306$,

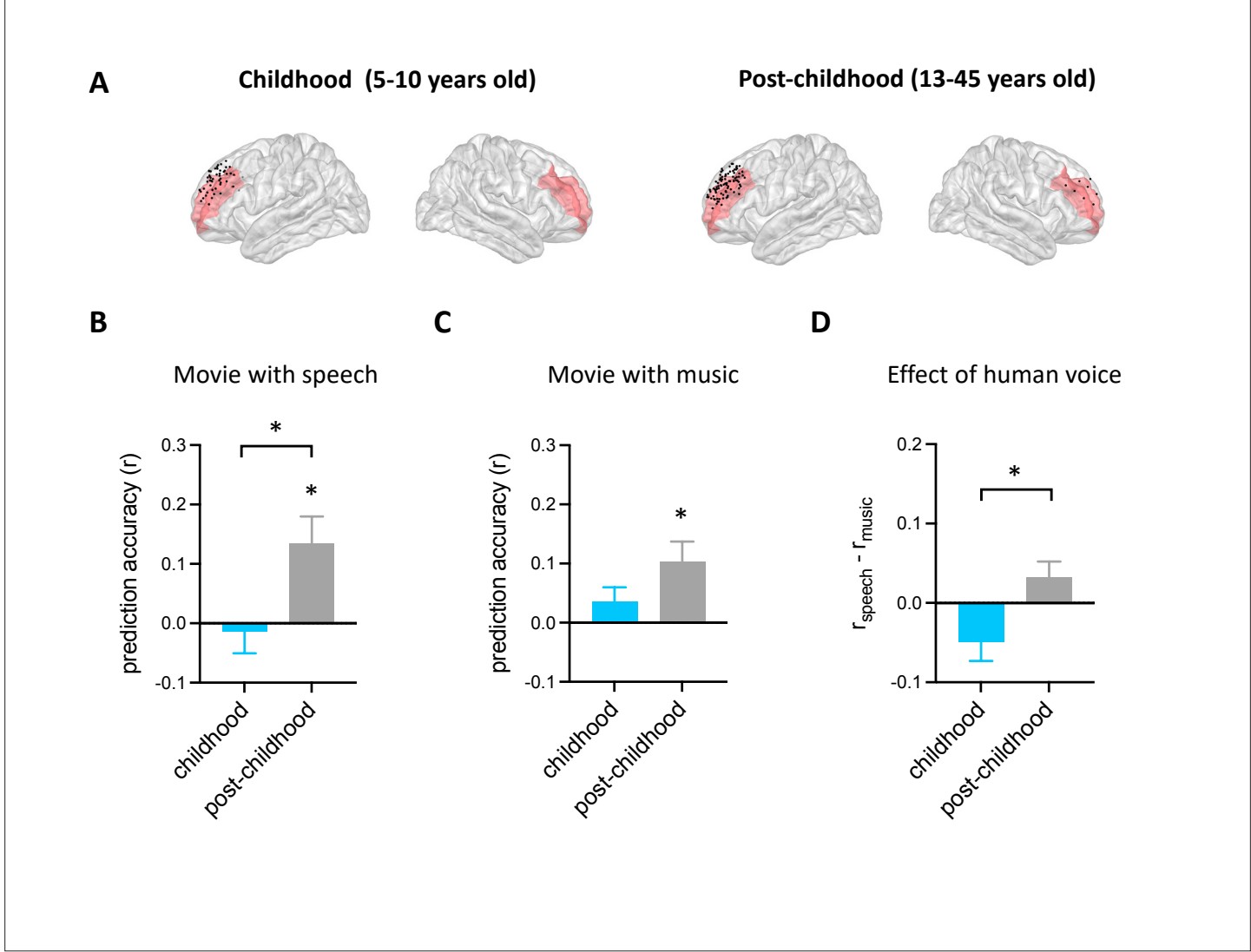

**Figure 2.** Prediction performance of encoding models in dorsolateral prefrontal cortex (DLPFC). (**A**) Spatial distribution of electrodes in DLPFC. Electrodes in all participants from each group are projected onto the Montreal Neurological Institute (MNI) space and shown on the average brain. Red shaded areas indicate middle frontal cortex provided by the FreeSurfer Desikan-Killiany atlas (**Desikan et al., 2006**). Electrodes outside the DLPFC are not shown. (**B**) Averaged prediction accuracy across participants for speech condition. The performance of the encoding model is measured as Pearson correlation coefficient (**r**) between measured and predicted brain activities. (**C**) Averaged prediction accuracy across participants for music condition. (**D**) Prediction accuracy difference between speech condition and music condition for each group. Error bars are standard error of the mean. *$p<0.05$. ($N_{childhood}=8$, $N_{post-childhood}=13$).

The online version of this article includes the following source data for figure 2:

**Source data 1.** Prediction performance of encoding models in dorsolateral prefrontal cortex (DLPFC).

two-tailed permutation test). In an example subject (S19: 8 years old) who has electrodes in both DLPFC and pSTC, the model fitting results clearly showed that facial emotion is encoded in pSTC (S19$_{speech}$: $p=0.0014$, $r=0.1951$), but not DLPFC (S19$_{speech}$: $p=0.295$, $r=-0.0628$). Similarly, group-level results showed that the model performance is significantly greater than zero in the pSTC of post-childhood individuals (**Figure 3D and E**, speech condition: $p=0.0001$, $t_{24}=4.601$, one-sample t-test), and this neural representation of facial expression information is significantly reduced when human voice is absent (**Figure 3E**, paired-t-test, $t_{24}=2.897$, $p=0.0079$). These results provide evidence that children's sensory areas encode facial emotion features in a manner similar to that of post-childhood individuals.

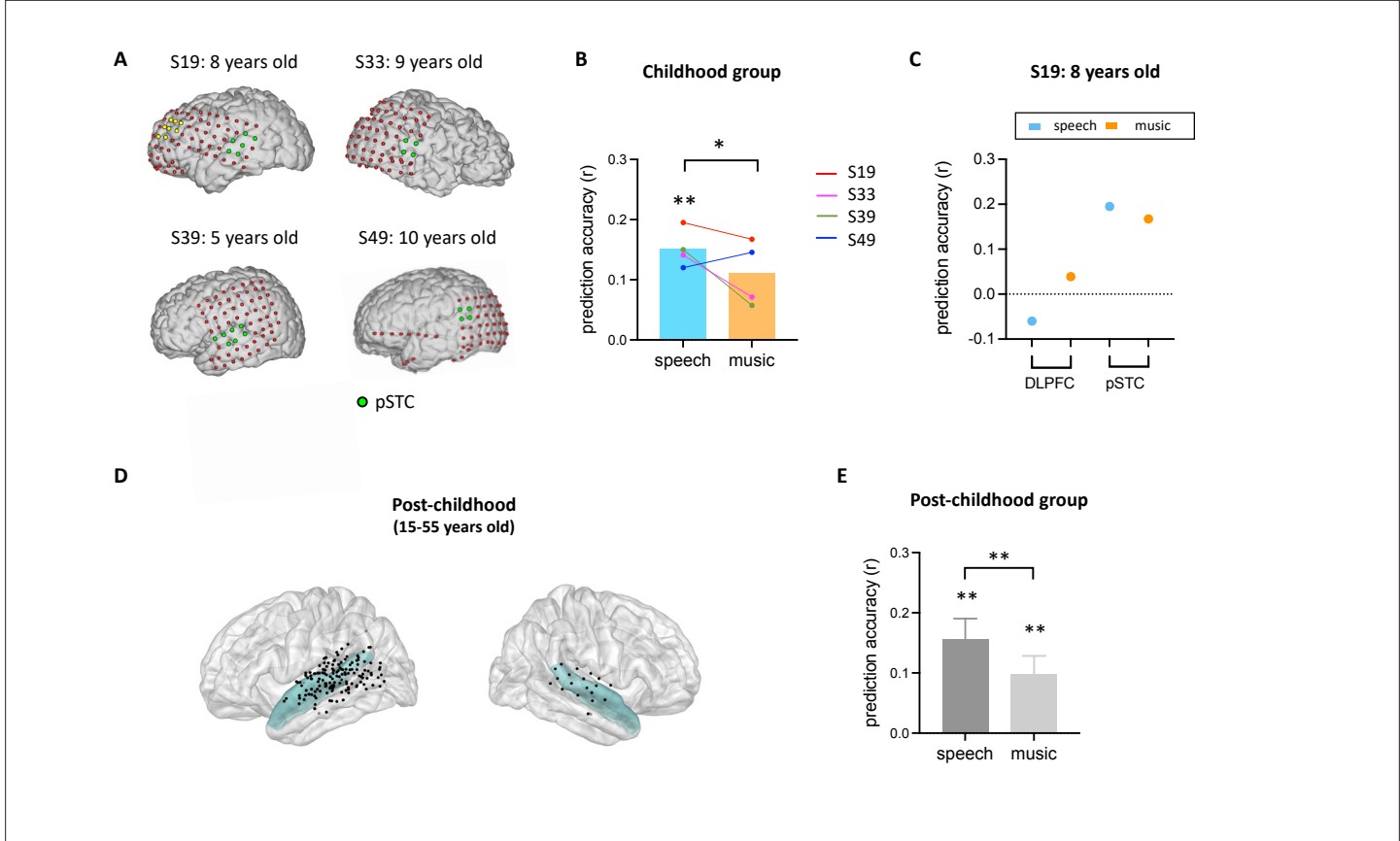

**Figure 3.** Prediction performance of encoding models in posterior superior temporal cortex (pSTC). (**A**) The electrode distribution in the native brain space for the four children. Electrodes in pSTC are green, and electrodes in dorsolateral prefrontal cortex (DLPFC) are yellow. (**B**) Prediction accuracy of encoding models in children. Bars indicate mean prediction accuracy across participants and lines indicate individual data. (**C**) Prediction accuracy of encoding models for S19 in both DLPFC and pSTC. (**D**) Spatial distribution of recording contacts in post-childhood participants' pSTC. The pSTC electrodes identified in individual space are projected onto Montreal Neurological Institute (MNI) space and shown on the average brain. Contacts other than pSTC are not shown. Blue shaded areas indicate superior temporal cortex provided by the FreeSurfer Desikan-Killiany atlas (**Desikan et al., 2006**). (**E**) Averaged prediction accuracy across post-childhood participants (N=25). Error bars are standard error of the mean. **$p<0.01$. *$p<0.05$.

The online version of this article includes the following source data for figure 3:

**Source data 1.** Prediction performance of encoding models in posterior superior temporal cortex (pSTC).

## The complexity of facial expression encoding in the pSTC increases across development

To understand how facial expression representation in pSTC changes across development, we examined the feature weights of the facial expression encoding models in all participants with significant prediction accuracy (10 post-childhood individuals and 2 children). The weight for each feature represents its relative contribution to predicting the neural response. First, we calculated the encoding weights for complex emotions (averaging guilt, embarrassment, pride, and envy, which were selected as the most representative complex emotions based on previous studies **Alba-Ferrara et al., 2011**; **Burnett et al., 2011**; **Russell and Paris, 1994**) and basic emotions (averaging joy, sadness, fear, anger, disgust, and surprise). Then, we calculated their correlations with age separately. Our results showed that the encoding weight of complex emotion was significantly positively correlated with age ($r_{12}=0.8512$, $p=0.004$, **Figure 4A** left). No significant correlation between encoding weight of basic emotion and age was observed ($r_{12}=0.3913$, $p=0.2085$, **Figure 4A** right). In addition, we computed Pearson correlations between each individual feature weight and age, ranking the r values from largest to smallest (**Figure 4B**). The highest correlations were found for embarrassment, guilt, pride, interest, and envy—emotions that are all considered complex emotions. Among them, the weights for embarrassment, guilt, pride, and interest showed significant positive correlations with

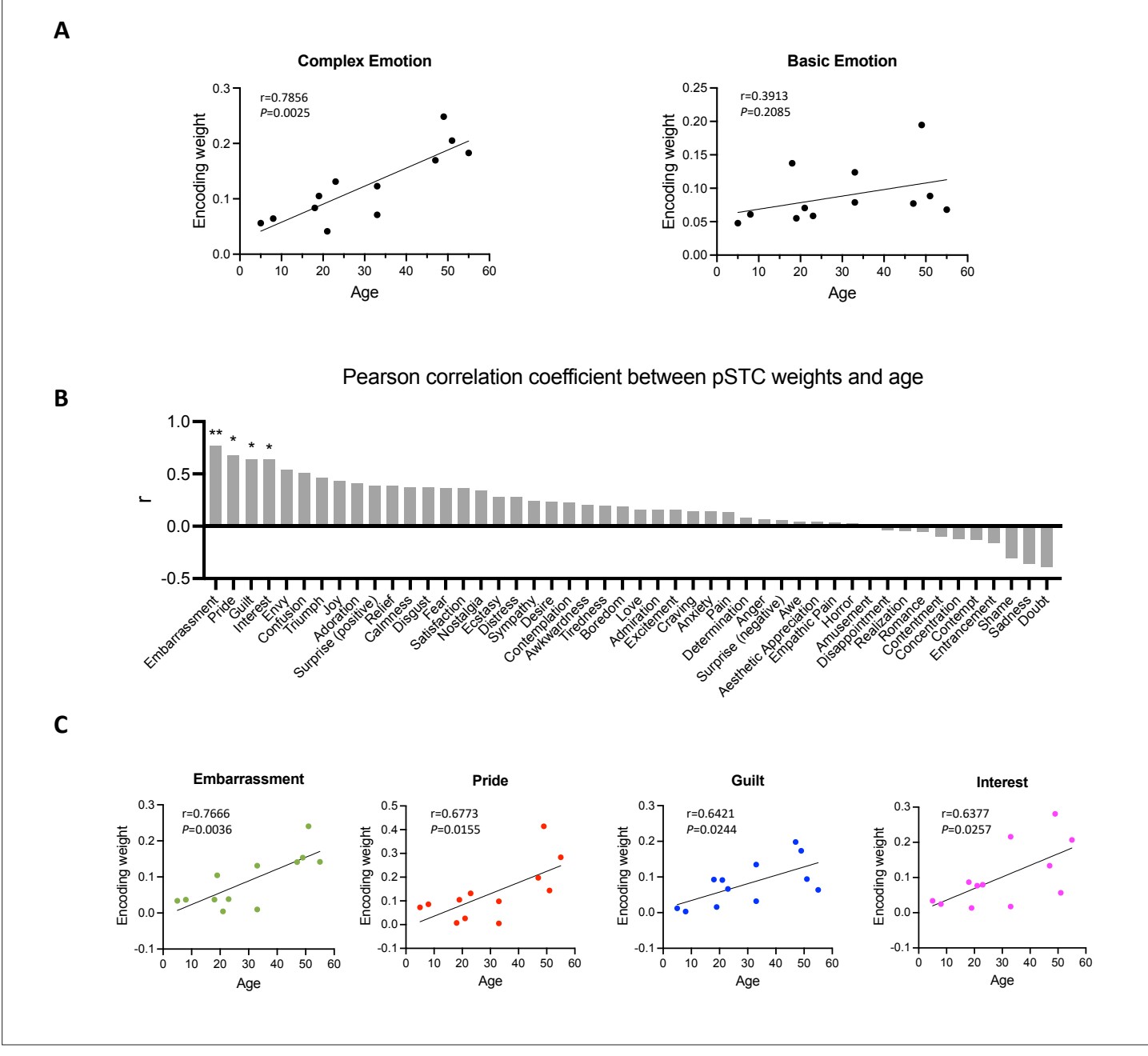

**Figure 4.** Correlation between encoding weights and age. (**A**) Left: Correlation between averaged encoding weights of five complex emotions and age. Right: Correlation between averaged encoding weights of six basic emotions and age. (**B**) Pearson correlation coefficient between encoding weights of 48 facial expression features and age. The results are ranked from largest to smallest. Significant correlations noted with * ($p<0.05$, uncorrected) or ** ($p<0.01$, uncorrected). (**C**) Correlation between encoding weights of embarrassment, pride, guilt, interest and age (N=12).

The online version of this article includes the following source data for figure 4:

**Source data 1.** Encoding weight and age.

age (*Figure 4C*, embarrassment: *r*=0.7666, *p*=0.0036; pride: *r*=0.6773, *p*=0.0155; guilt: *r*=0.6421, *p*=0.0244, interest: *r*=0.6377, *p*=0.0257, uncorrected for multiple comparisons), suggesting that the encoding of these complex emotions in pSTC increases with age. Thus, our results suggest that as development progresses, the pSTC becomes increasingly engaged in encoding complex emotions which requires representing others' mental states and emerges later in development (*Burnett et al., 2009*; *Garcia and Scherf, 2015*; *Motta-Mena and Scherf, 2017*).

## Discussion

The current study examines functional changes in both low-level and high-level brain areas across development to provide valuable insights into the neural mechanisms underlying the maturation of facial expression perception. Based on our findings, we propose that young children rely primarily on early sensory areas rather than the prefrontal cortex for facial emotion processing. As development progresses, the prefrontal cortex becomes increasingly involved, perhaps serving to modulate responses in early sensory areas based on emotional context and enabling them to process complex emotions. This developmental progression ultimately enables the full comprehension of facial emotions in adulthood.

Behavioral results suggest that infants as young as only 7–8 months can categorize some emotions (*Caron et al., 1982*; *Nelson and Dolgin, 1985*; *Nelson et al., 1979*). However, sensitivity to facial expressions in young children does not mean that they can understand the meaning of that affective state. For example, *Kaneshige and Haryu, 2015* found that although 4-month-old infants could discriminate facial configurations of anger and happiness, they responded positively to both, suggesting that at this stage, they may lack knowledge of the affective meaning behind these expressions. This underscores the idea that additional processes need to be developed for children to fully grasp the emotional content conveyed by facial expressions. Although the neural mechanism behind this development is still unclear, a reasonable perspective is that it requires both visual processing of facial features and emotion-related processing for the awareness of the emotional state of the other person (*Pereira et al., 2019*; *Pollak et al., 2009*; *Ruba and Pollak, 2020*). Indeed, growing evidence suggests that the prefrontal cortex plays an important role in integrating prior knowledge with incoming sensory information, allowing interpretation of the current situation in light of past emotional experience (*Batty and Taylor, 2006*; *Williams et al., 2006*).

In the current study, we observed differential representation of facial expressions in the DLPFC between children and post-childhood individuals. First, in post-childhood individuals, neural activity in the DLPFC encodes high-dimensional facial expression information, whereas this encoding is absent in children. Second, while human voice enhances the representation of facial expressions in the DLPFC of post-childhood individuals, it instead reduces this representation in children. These results suggest that the DLPFC undergoes developmental changes in how it processes facial expressions. The absence of high-dimensional facial expression encoding in children implies that the DLPFC may not yet be fully engaged in emotional interpretation at an early age. Additionally, the opposite effects of human voice on facial expression representation indicate that multimodal integration of social cues develops over time. In post-childhood individuals, voices may enhance emotional processing by providing congruent information (*Collignon et al., 2008*; *Klasen et al., 2012*; *Kreifelts et al., 2007*), whereas in children, the presence of voice might interfere with or redirect attentional resources away from facial expression processing (*Jacob et al., 2014*; *Klasen et al., 2012*; *Weissman et al., 2004*).

There have been few neuroimaging studies directly examining the functional role of young children's DLPFC in facial emotion perception. Some evidence suggests that the prefrontal cortex continues to develop until adulthood to achieve its mature function in emotion perception (*Gunning-Dixon et al., 2003*; *Tessitore et al., 2005*; *Thomas et al., 2007*), and for some emotion categories, this development may extend across the lifespan. For example, prefrontal cortex activation during viewing fearful faces increases with age (*Monk et al., 2003*; *Williams et al., 2006*; *Yurgelun-Todd and Killgore, 2006*). As there were not enough participants for us to calculate correlation between encoding model performance in DLPFC and age, it is still unclear whether the representation of facial expression in DLPFC increases linearly with age. One possibility is that the representation of facial expressions in the DLPFC gradually increases with age until it reaches an adult-like level. This would suggest a continuous developmental trajectory, where incremental improvements in neural processing accumulate over time. Another possibility is that development follows a more nonlinear pattern, showing improvement with prominent changes at specific ages. Interestingly, research has shown that performance on matching emotional expressions improves steadily over development, with notable gains in accuracy occurring between 9 and 10 years and again between 13 and 14 years, after which performance reaches adult-like levels (*Kolb et al., 1992*).

Our results clearly showed that facial expression is encoded in children's pSTC. Childhood and the post-childhood group showed comparable levels of prediction accuracy. In the 8-year-old child (S19) who had electrode coverage in both DLPFC and pSTC, facial expressions were represented in

the pSTC but not in the DLPFC. This rare sampling allows us to rule out the possibility that the low prediction accuracy of the facial expression encoding model in the DLPFC is due to the reduced engagement in the movie-watching task for children. Consistent with our findings, previous studies have shown that the fusiform and superior temporal gyri are involved in emotion-specific processing in 10-year-old children (*Lobaugh et al., 2006*). Meanwhile, some other researchers found that responses to facial expression in the amygdala and posterior fusiform gyri decreased as people got older (*Tessitore et al., 2005*), but the use of frontal regions increased with age (*Gunning-Dixon et al., 2003*). Therefore, we propose that early sensory areas like the fusiform and superior temporal gyri play a key role in facial expression processing in children, but their contribution may shift with age as frontal regions become more involved. Consistent with this perspective, our results revealed that the encoding weights for complex emotions in pSTC increased with age, implying a developmental trajectory in the neural representation of complex emotions in pSTC. This finding aligns with previous behavioral studies showing that social complex emotion recognition does not fully mature until young adulthood (*Burnett et al., 2009*; *Garcia and Scherf, 2015*; *Motta-Mena and Scherf, 2017*). In fact, our results imply that the representation of complex facial expressions in pSTC continues to develop over the lifespan. As for the correlation between basic emotion encoding and age, the lack of a significant effect in our study does not necessarily indicate an absence of developmental change but may instead be due to the limited sample size.

Our study examines how facial emotions presented in naturalistic video stimuli are represented in the human brain across developmental stages. The facial emotional features used in our encoding models describe the emotions portrayed by characters in the stimulus, rather than participants' subjective perception or experience of those emotions. While our results suggest the potential neural basis for participants' perception of these emotions, we cannot conclude that such information was consciously perceived without direct behavioral measures. In fact, automatic and efficient emotion perception is not driven by the structural features of a face alone but is also shaped by other factors, such as contextual information, prior experiences, and current mood states (*Barrett et al., 2011*). How the conscious percept of facial emotion is constructed remains far from fully understood. Recent evidence suggests that dynamic interactions between visual cortices and frontal regions integrate stimulus-defined features with contextual and internal factors to give rise to subjective emotional perception (*Wang et al., 2014*).

## Limitations and future directions

Our study provides novel insights into the neural mechanisms underlying the development of facial expression processing. As with any study, several limitations should be acknowledged. First, most electrode coverage in our study was in the left hemisphere, potentially limiting our understanding of lateralization effects. Second, while our results provide insights into the role of DLPFC during development, we were unable to examine other prefrontal regions, such as the orbitofrontal cortex (OFC) and anterior cingulate cortex (ACC), to examine their unique contributions to emotion processing. Third, due to sample size constraints, we were unable to divide participants into more granular developmental stages, such as early childhood, adolescence, and adulthood, which could provide a more detailed characterization of the neural mechanisms underlying the development of facial expression processing. Future studies using non-invasive methods with more age-diverse samples, or with longitudinal designs to directly track developmental trajectories, will be essential for refining our understanding of how facial emotion processing develops across the lifespan. In addition, future studies are needed to systematically examine how multimodal cues and prior experiences contribute to the understanding of emotion from faces.

## Methods

### Key resources table

| Reagent type (species) or resource | Designation | Source or reference | Identifiers | Additional information |
|---|---|---|---|---|
| Software, algorithm | MNE-Python | MNE-Python | RRID:SCR_005972 | |
| Software, algorithm | Python | Python | RRID:SCR_008394 | |

*Continued on next page*

*Continued*

| Reagent type (species) or resource | Designation | Source or reference | Identifiers | Additional information |
|---|---|---|---|---|
| Software, algorithm | FreeSurfer | FreeSurfer | RRID:SCR_001847 | |
| Software, algorithm | Hume AI | Hume AI | | |

In this study, iEEG data from an open multimodal iEEG-fMRI dataset were analyzed (*Berezutskaya et al., 2022*).

### Participants and electrode distribution

Due to the research purposes of the current study, only participants who had at least four electrode contacts in either DLPFC or pSTC were included in the data analysis (*Tables 1 and 2*). Eleven children (5–10 years old, 7 females) and thirty-one post-childhood individuals (13–55 years old, 18 females) are included in the present study. In the childhood group, eight participants had enough electrodes implanted in the DLPFC, and four had enough electrodes implanted in the pSTC. In the post-childhood group, thirteen participants had enough electrodes implanted in the DLPFC, and twenty-five had enough electrodes implanted in the pSTC.

### Experimental procedures

A 6.5 min short film was crafted by editing fragments from *Pippi on the Run* into a coherent narrative. The film is structured into 13 interleaved 30 s blocks of either speech or music, with seven blocks featuring background music only and six blocks retaining the original dialogue and voice from the video. Patients were asked to watch the movie while the intracranial EEG signals were recorded. No fixation cross was displayed in the middle of the screen or elsewhere. The movie was presented using the Presentation software (Neurobehavioral Systems, Berkeley, CA) and the sound was synchronized with the neural recordings. All patients were asked to participate in the experiment outside the window of seizures. More data acquisition details can be found in *Berezutskaya et al., 2022*.

### iEEG data processing

Electrode contacts and epochs contaminated with excessive artifacts and epileptiform activity were removed from data analysis by visual inspection. Raw data were filtered with a 50 Hz notch filter and re-referenced to the common average reference. For each electrode contact in each patient, the preprocessed data were band-pass filtered (110–140 Hz, fourth-order Butterworth). The Hilbert

**Table 1.** Demographic information of childhood group.

| ID | Sex | Age | Number of recording contacts in left pSTC | Number of recording contacts in right pSTC | Number of recording contacts in left DLPFC | Number of recording contacts in right DLPFC |
|---|---|---|---|---|---|---|
| s02 | F | 9 | 0 | 0 | 9 | 0 |
| s10 | F | 8 | 0 | 0 | 8 | 0 |
| s19 | F | 8 | 6 | 0 | 8 | 0 |
| s32 | F | 6 | 0 | 0 | 5 | 0 |
| s33 | F | 9 | 0 | 4 | 0 | 0 |
| s37 | M | 5 | 0 | 0 | 4 | 0 |
| s39 | F | 5 | 7 | 0 | 0 | 0 |
| s41 | M | 7 | 0 | 0 | 4 | 0 |
| s49 | F | 10 | 4 | 0 | 0 | 0 |
| s50 | F | 9 | 0 | 0 | 6 | 0 |
| s63 | M | 5 | 0 | 0 | 9 | 0 |

Table 2. Demographic information of post-childhood group.

| ID | Sex | Age | Number of recording contacts in left pSTC | Number of recording contacts in right pSTC | Number of recording contacts in left DLPFC | Number of recording contacts in right DLPFC |
|---|---|---|---|---|---|---|
| s1 | M | 55 | 0 | 4 | 0 | 0 |
| s3 | F | 33 | 6 | 0 | 6 | 0 |
| s5 | F | 33 | 8 | 0 | 8 | 0 |
| s6 | F | 43 | 9 | 0 | 0 | 0 |
| s12 | M | 37 | 0 | 0 | 12 | 0 |
| s14 | F | 18 | 4 | 0 | 8 | 0 |
| s16 | M | 17 | 8 | 0 | 9 | 0 |
| s17 | M | 28 | 0 | 8 | 0 | 0 |
| s18 | F | 15 | 6 | 0 | 11 | 0 |
| s20 | F | 25 | 9 | 0 | 0 | 0 |
| s22 | M | 21 | 4 | 0 | 0 | 0 |
| s24 | F | 47 | 11 | 0 | 0 | 0 |
| s25 | M | 14 | 0 | 0 | 6 | 0 |
| s26 | F | 48 | 11 | 0 | 0 | 0 |
| s27 | M | 15 | 11 | 0 | 0 | 0 |
| s28 | M | 21 | 0 | 6 | 0 | 0 |
| s31 | F | 13 | 0 | 0 | 7 | 0 |
| s34 | F | 51 | 5 | 0 | 0 | 0 |
| s38 | F | 14 | 0 | 0 | 4 | 0 |
| s40 | M | 49 | 6 | 0 | 0 | 0 |
| s43 | M | 19 | 0 | 4 | 0 | 8 |
| s45 | M | 19 | 0 | 0 | 4 | 0 |
| s48 | F | 18 | 5 | 0 | 0 | 0 |
| s51 | M | 46 | 6 | 0 | 9 | 0 |
| s54 | F | 31 | 4 | 0 | 0 | 0 |
| s55 | F | 23 | 6 | 0 | 0 | 0 |
| s57 | F | 36 | 0 | 0 | 7 | 0 |
| s58 | F | 16 | 6 | 0 | 0 | 0 |
| s59 | F | 30 | 7 | 0 | 0 | 0 |
| s60 | M | 42 | 8 | 0 | 0 | 0 |
| s61 | F | 16 | 8 | 0 | 0 | 0 |

transform was then applied to extract the analytic amplitude. Each event (block) was extracted in the 0–30 s time window around its onset. The fifth music block was excluded, as there were no faces presented on screen. Subsequently, the data were down-sampled to 400 Hz and square-root transformed. Finally, the data were normalized by z-scoring with respect to baseline periods (−0.2–0 s before stimulus onset).

## Contact location and regions of interest

We identified electrode contacts in STC in individual brains using individual anatomical landmarks (i.e. gyri and sulci). Superior temporal sulci and lateral sulci were used as boundaries. A coronal plane, including the posterior tip of the hippocampus served as an anterior/posterior boundary. To identify electrode contacts in DLPFC, we projected the electrode contact positions provided by the open dataset onto the Montreal Neurological Institute-152 template brain (MNI) space, using FreeSurfer. DLPFC was defined based on the following sets of HCP-MMP1 (*Glasser et al., 2016*) labels on both left and right hemispheres: 9-46d, 46, a9-46v, and p9-46v.

## Emotion feature extraction

Hume AI (https://www.hume.ai) was used to extract the facial emotion features from the video. When multiple faces appeared in the movie, the maximum score of the facial expression features across all faces was used for each emotion category. All the time courses of facial emotion features were resampled to 2 Hz. No facial emotion features were extracted for the fifth music block due to the absence of faces. The full list of the 48 facial emotion features is shown in *Figure 4B*.

## Encoding model fitting

To model iEEG responses to emotion, we used a linear regression approach with 48 facial emotion features extracted by Hume AI. Time-lagged versions of each feature (with 0, 0.5, and 1 s delays) were used in the model fitting. For each participant, high-frequency broadband (HFB) responses from all electrode contacts within each area were concatenated. To match the temporal resolution of the emotion feature time course, the HFB responses were binned into 500 ms windows. We then modeled the processed HFB response for each participant, each brain area, and each condition (speech vs music) using ridge regression. The optimal regularization parameter was assessed using fivefold cross-validation, with the 20 different regularization parameters (log spaced between 10 and 10000). To keep the scale of the weights consistent, a single best overall value of the regularization coefficient was used for all areas in both the speech and music conditions in all patients. We used a cross-validation iterator to fit the model and test it on held-out data. The model performance was evaluated by calculating Pearson correlation coefficients between measured and predicted HFB response of individual brain areas. The mean prediction accuracy (r value) of the encoding model with fivefold cross-validation was then calculated. A one-sample t-test was used to test whether the encoding model performance was significantly >0. For children's pSTC, non-parametric permutation tests were used to test whether the encoding model performance was significantly >0 and whether there was a significant difference between groups. Specifically, we shuffled facial emotion feature data in time, and then we conducted the standard data analysis steps (described above) using the shuffled facial emotion features. This shuffle procedure was repeated 5000 times to generate a null distribution, and p-values were calculated as the proportion of results from shuffled data more extreme than the observed real value. A two-sided paired t-test was used to examine differences in encoding accuracy between speech and music conditions in the post-childhood group.

## Weight analysis

To examine the correlation between encoding model weights and age, we obtained 48 encoding model weights from all folds of cross-validation for all participants whose pSTC significantly encoded facial expression (i.e. the p-value of prediction accuracy is less than 0.05). Thus, 10 post-childhood individuals and 2 children were involved in the weight analysis. The weight for each feature represents its relative contribution to predicting the neural response. A higher weight indicates that the corresponding feature has a stronger influence on neural activity, meaning that variations in this feature more significantly impact the predicted response. We used the absolute value of weights and, therefore, did not discriminate whether facial emotion features were mapped to an increase or decrease in the HFB response.

## Acknowledgements

We would like to thank Dr. Julia Berezutskaya for providing the audiovisual film that was used for iEEG data collection. This work was supported by funding from the United States National Institutes of Health (R01-MH127006).

## Additional information

### Funding

| Funder | Grant reference number | Author |
|---|---|---|
| National Institute of Mental Health | R01-MH127006 | Kelly Bijanki |

The funders had no role in study design, data collection and interpretation, or the decision to submit the work for publication.

### Author contributions

Xiaoxu Fan, Conceptualization, Formal analysis, Methodology, Writing - original draft, Writing – review and editing; Abhishek Tripathi, Formal analysis; Kelly Bijanki, Funding acquisition, Project administration, Writing – review and editing

### Author ORCIDs

Xiaoxu Fan ⓘ https://orcid.org/0000-0002-8115-8621
Kelly Bijanki ⓘ https://orcid.org/0000-0003-1624-8767

Reviewer #1 (Public review): https://doi.org/10.7554/eLife.107636.3.sa1
Reviewer #2 (Public review): https://doi.org/10.7554/eLife.107636.3.sa2
Author response https://doi.org/10.7554/eLife.107636.3.sa3

## Additional files

### Supplementary files

MDAR checklist

### Data availability

iEEG data is available in openneuro. Source code is available at OSF (https://osf.io/d2th6). Figure 2-source data 1, Figure 3-source data 1 and Figure 4-source data 1 contain the numerical data used to generate the figures.

The following dataset was generated:

| Author(s) | Year | Dataset title | Dataset URL | Database and Identifier |
|---|---|---|---|---|
| Fan X | 2025 | The representation of facial emotion expands from sensory to prefrontal cortex with development | https://osf.io/d2th6 | Open Science Framework, d2th6 |

The following previously published dataset was used:

| Author(s) | Year | Dataset title | Dataset URL | Database and Identifier |
|---|---|---|---|---|
| Berezutskaya J, Vansteensel MJ, Aarnoutse EJ, Freudenburg ZV, Piantoni G, Branco MP, Ramsey NF | 2022 | Open multimodal iEEG-fMRI dataset from naturalistic stimulation with a short audiovisual film | https://doi.org/10.18112/openneuro.ds003688.v1.0.7 | OpenNeuro, 10.18112/openneuro.ds003688.v1.0.7 |

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
