## [Editor Report · eLife Assessment]

This study examines an **important** question regarding the developmental trajectory of neural mechanisms supporting facial expression processing. Leveraging a rare intracranial EEG (iEEG) dataset including both children and adults, the authors reported that facial expression recognition mainly engaged the posterior superior temporal cortex (pSTC) among children, while both pSTC and the prefrontal cortex were engaged among adults. In terms of strength of evidence, the **solid** methods, data and analyses broadly support the claims with minor weaknesses.

---

## [Referee Report · Reviewer #1 (Public review)]

Summary:

This study investigates how the brain processes facial expressions across development by analyzing intracranial EEG (iEEG) data from children (ages 5-10) and post-childhood individuals (ages 13-55). The researchers used a short film containing emotional facial expressions and applied AI-based models to decode brain responses to facial emotions. They found that in children, facial emotion information is represented primarily in the posterior superior temporal cortex (pSTC)-a sensory processing area-but not in the dorsolateral prefrontal cortex (DLPFC), which is involved in higher-level social cognition. In contrast, post-childhood individuals showed emotion encoding in both regions. Importantly, the complexity of emotions encoded in the pSTC increased with age, particularly for socially nuanced emotions like embarrassment, guilt, and pride.The authors claim that these findings suggest that emotion recognition matures through increasing involvement of the prefrontal cortex, supporting a developmental trajectory where top-down modulation enhances understanding of complex emotions as children grow older.

Strengths:

(1) The inclusion of pediatric iEEG makes this study uniquely positioned to offer high-resolution temporal and spatial insights into neural development compared to non-invasive approaches, e.g., fMRI, scalp EEG, etc.

(2) Using a naturalistic film paradigm enhances ecological validity compared to static image tasks often used in emotion studies.

(3) The idea of using state-of-the-art AI models to extract facial emotion features allows for high-dimensional and dynamic emotion labeling in real time.

Weaknesses:

(1) The study has notable limitations that constrain the generalizability and depth of its conclusions. The sample size was very small, with only nine children included and just two having sufficient electrode coverage in the posterior superior temporal cortex (pSTC), which weakens the reliability and statistical power of the findings, especially for analyses involving age. Authors pointed out that a similar sample size has been used in previous iEEG studies, but the cited works focus on adults and do not look at the developmental perspectives. Similar work looking at developmental changes in iEEG signals usually includes many more subjects (e.g., n = 101 children from Cross ZR et al., Nature Human Behavior, 2025) to account for inter-subject variabilities.

(2) Electrode coverage was also uneven across brain regions, with not all participants having electrodes in both the dorsolateral prefrontal cortex (DLPFC) and pSTC, making the conclusion regarding the different developmental changes between DLPFC and pSTC hard to interpret (related to point 3 below). It is understood that it is rare to have such iEEG data collected in this age group, and the electrode location is only determined by clinical needs. However, the scientific rigor should not be compromised by the limited data access. It's the authors' decision whether such an approach is valid and appropriate to address the scientific questions, here the developmental changes in the brain, given all the advantages and constraints of the data modality.

(3) The developmental differences observed were based on cross-sectional comparisons rather than longitudinal data, reducing the ability to draw causal conclusions about developmental trajectories. Also, see comments in point 2.

(4) Moreover, the analysis focused narrowly on DLPFC, neglecting other relevant prefrontal areas such as the orbitofrontal cortex (OFC) and anterior cingulate cortex (ACC), which play key roles in emotion and social processing. Agree that this might be beyond the scope of this paper, but a discussion section might be insightful.

(5) Although the use of a naturalistic film stimulus enhances ecological validity, it comes at the cost of experimental control, with no behavioral confirmation of the emotions perceived by participants and uncertain model validity for complex emotional expressions in children. A non-facial music block that could have served as a control was available but not analyzed. The validation of AI model's emotional output needs to be tested. It is understood that we cannot collect these behavioral data retrospectively within the recorded subjects. Maybe potential post-hoc experiments and analyses could be done, e.g., collect behavioral, emotional perception data from age-matched healthy subjects.

(6) Generalizability is further limited by the fact that all participants were neurosurgical patients, potentially with neurological conditions such as epilepsy that may influence brain responses. At least some behavioral measures between the patient population and the healthy groups should be done to ensure the perception of emotions is similar.

(7) Additionally, the high temporal resolution of intracranial EEG was not fully utilized, as data were downsampled and averaged in 500-ms windows. It seems like the authors are trying to compromise the iEEG data analyses to match up with the AI's output resolution, which is 2Hz. It is not clear then why not directly use fMRI, which is non-invasive and seems to meet the needs here already. The advantages of using iEEG in this study are missing here.

(8) Finally, the absence of behavioral measures or eye-tracking data makes it difficult to directly link neural activity to emotional understanding or determine which facial features participants attended to. Related to point 5 as well.

Comments on revisions:

A behavioral measurement will help address a lot of these questions. If the data continues collecting, additional subjects with iEEG recording and also behavioral measurements would be valuable.

---

## [Referee Report · Reviewer #2 (Public review)]

Summary:

In this paper, Fan et al. aim to characterize how neural representations of facial emotions evolve from childhood to adulthood. Using intracranial EEG recordings from participants aged 5 to 55, the authors assess the encoding of emotional content in high-level cortical regions. They report that while both the posterior superior temporal cortex (pSTC) and dorsolateral prefrontal cortex (DLPFC) are involved in representing facial emotions in older individuals, only the pSTC shows significant encoding in children. Moreover, the encoding of complex emotions in the pSTC appears to strengthen with age. These findings lead the authors to suggest that young children rely more on low-level sensory areas and propose a developmental shift from reliance on lower-level sensory areas in early childhood to increased top-down modulation by the prefrontal cortex as individuals mature.

Strengths:

(1) Rare and valuable dataset: The use of intracranial EEG recordings in a developmental sample is highly unusual and provides a unique opportunity to investigate neural dynamics with both high spatial and temporal resolution.

(2) Developmentally relevant design: The broad age range and cross-sectional design are well-suited to explore age-related changes in neural representations.

(3) Ecological validity: The use of naturalistic stimuli (movie clips) increases the ecological relevance of the findings.

(4) Feature-based analysis: The authors employ AI-based tools to extract emotion-related features from naturalistic stimuli, which enables a data-driven approach to decoding neural representations of emotional content. This method allows for a more fine-grained analysis of emotion processing beyond traditional categorical labels.

Weaknesses:

(1) While the authors leverage Hume AI, a tool pre-trained on a large dataset, its specific performance on the stimuli used in this study remains unverified. To strengthen the foundation of the analysis, it would be important to confirm that Hume AI's emotional classifications align with human perception for these particular videos. A straightforward way to address this would be to recruit human raters to evaluate the emotional content of the stimuli and compare their ratings to the model's outputs.

(2) Although the study includes data from four children with pSTC coverage-an increase from the initial submission-the sample size remains modest compared to recent iEEG studies in the field.

(3) The "post-childhood" group (ages 13-55) conflates several distinct neurodevelopmental periods, including adolescence, young adulthood, and middle adulthood. As a finer age stratification is likely not feasible with the current sample size, I would suggest authors temper their developmental conclusions.

(4) The analysis of DLPFC-pSTC directional connectivity would be significantly strengthened by modeling it as a continuous function of age across all participants, rather than relying on an unbalanced comparison between a single child and a (N=7) post-childhood group. This continuous approach would provide a more powerful and nuanced view of the developmental trajectory. I would also suggest including the result in the main text.

---

## [Author Response]

The following is the authors’ response to the original reviews.

**eLife Assessment**
This study examines a valuable question regarding the developmental trajectory of neural mechanisms supporting facial expression processing. Leveraging a rare intracranial EEG (iEEG) dataset including both children and adults, the authors reported that facial expression recognition mainly engaged the posterior superior temporal cortex (pSTC) among children, while both pSTC and the prefrontal cortex were engaged among adults. However, the sample size is relatively small, with analyses appearing incomplete to fully support the primary claims.
**Public Reviews:**

**Reviewer #1 (Public review):**
Summary:This study investigates how the brain processes facial expressions across development by analyzing intracranial EEG (iEEG) data from children (ages 5-10) and post-childhood individuals (ages 13-55). The researchers used a short film containing emotional facial expressions and applied AI-based models to decode brain responses to facial emotions. They found that in children, facial emotion information is represented primarily in the posterior superior temporal cortex (pSTC) - a sensory processing area - but not in the dorsolateral prefrontal cortex (DLPFC), which is involved in higher-level social cognition. In contrast, post-childhood individuals showed emotion encoding in both regions. Importantly, the complexity of emotions encoded in the pSTC increased with age, particularly for socially nuanced emotions like embarrassment, guilt, and pride. The authors claim that these findings suggest that emotion recognition matures through increasing involvement of the prefrontal cortex, supporting a developmental trajectory where top-down modulation enhances understanding of complex emotions as children grow older.Strengths:(1) The inclusion of pediatric iEEG makes this study uniquely positioned to offer high-resolution temporal and spatial insights into neural development compared to non-invasive approaches, e.g., fMRI, scalp EEG, etc.(2) Using a naturalistic film paradigm enhances ecological validity compared to static image tasks often used in emotion studies.(3) The idea of using state-of-the-art AI models to extract facial emotion features allows for high-dimensional and dynamic emotion labeling in real timeWeaknesses:(1) The study has notable limitations that constrain the generalizability and depth of its conclusions. The sample size was very small, with only nine children included and just two having sufficient electrode coverage in the posterior superior temporal cortex (pSTC), which weakens the reliability and statistical power of the findings, especially for analyses involving age

We appreciated the reviewer’s point regarding the constrained sample size.

As an invasive method, iEEG recordings can only be obtained from patients undergoing electrode implantation for clinical purposes. Thus, iEEG data from young children are extremely rare, and rapidly increasing the sample size within a few years is not feasible. However, we are confident in the reliability of our main conclusions. Specifically, 8 children (53 recording contacts in total) and 13 control participants (99 recording contacts in total) with electrode coverage in the DLPFC are included in our DLPFC analysis. This sample size is comparable to other iEEG studies with similar experiment designs [1-3].

For pSTC, we returned to the data set and found another two children who had pSTC coverage. After involving these children’s data, the group-level analysis using permutation test showed that children’s pSTC significantly encode facial emotion in naturalistic contexts (Figure 3B). Notably, the two new children’s (S33 and S49) responses were highly consistent with our previous observations. Moreover, the averaged prediction accuracy in children’s pSTC (r_speech_=0.1565) was highly comparable to that in post-childhood group (r_speech_=0.1515).

(1) Zheng, J. et al. Multiplexing of Theta and Alpha Rhythms in the Amygdala-Hippocampal Circuit Supports Pafern Separation of Emotional Information. Neuron 102, 887-898.e5 (2019).

(2) Diamond, J. M. et al. Focal seizures induce spatiotemporally organized spiking activity in the human cortex. Nat. Commun. 15, 7075 (2024).

(3) Schrouff, J. et al. Fast temporal dynamics and causal relevance of face processing in the human temporal cortex. Nat. Commun. 11, 656 (2020).

(2) Electrode coverage was also uneven across brain regions, with not all participants having electrodes in both the dorsolateral prefrontal cortex (DLPFC) and pSTC, and most coverage limited to the left hemisphere-hindering within-subject comparisons and limiting insights into lateralization.

The electrode coverage in each patient is determined entirely by the clinical needs. Only a few patients have electrodes in both DLPFC and pSTC because these two regions are far apart, so it’s rare for a single patient’s suspected seizure network to span such a large territory. However, it does not affect our results, as most iEEG studies combine data from multiple patients to achieve sufficient electrode coverage in each target brain area. As our data are mainly from left hemisphere (due to the clinical needs), this study was not designed to examine whether there is a difference between hemispheres in emotion encoding. Nevertheless, lateralization remains an interesting question that should be addressed in future research, and we have noted this limitation in the Discussion (Page 8, in the last paragraph of the Discussion).

(3) The developmental differences observed were based on cross-sectional comparisons rather than longitudinal data, reducing the ability to draw causal conclusions about developmental trajectories.

In the context of pediatric intracranial EEG, longitudinal data collection is not feasible due to the invasive nature of electrode implantation. We have added this point to the Discussion to acknowledge that while our results reveal robust age-related differences in the cortical encoding of facial emotions, longitudinal studies using non-invasive methods will be essential to directly track developmental trajectories (Page 8, in the last paragraph of Discussion). In addition, we revised our manuscript to avoid emphasis causal conclusions about developmental trajectories in the current study (For example, we use “imply” instead of “suggest” in the fifth paragraph of Discussion).

(4) Moreover, the analysis focused narrowly on DLPFC, neglecting other relevant prefrontal areas such as the orbitofrontal cortex (OFC) and anterior cingulate cortex (ACC), which play key roles in emotion and social processing.

We agree that both OFC and ACC are critically involved in emotion and social processing. However, we have no recordings from these areas because ECoG rarely covers the ACC or OFC due to technical constraints. We have noted this limitation in the Discussion(Page 8, in the last paragraph of Discussion). Future follow-up studies using sEEG or non-invasive imaging methods could be used to examine developmental patterns in these regions.

(5) Although the use of a naturalistic film stimulus enhances ecological validity, it comes at the cost of experimental control, with no behavioral confirmation of the emotions perceived by participants and uncertain model validity for complex emotional expressions in children. A nonfacial music block that could have served as a control was available but not analyzed.

The facial emotion features used in our encoding models were extracted by Hume AI models, which were trained on human intensity ratings of large-scale, experimentally controlled emotional expression data [1-2]. Thus, the outputs of Hume AI model reflect what typical facial expressions convey, that is, the presented facial emotion. Our goal of the present study was to examine how facial emotions presented in the videos are encoded in the human brain at different developmental stages. We agree that children’s interpretation of complex emotions may differ from that of adults, resulting in different perceived emotion (i.e., the emotion that the observer subjectively interprets). Behavioral ratings are necessary to study the encoding of subjectively perceived emotion, which is a very interesting direction but beyond the scope of the present work. We have added a paragraph in the Discussion (see Page 8) to explicitly note that our study focused on the encoding of presented emotion.

We appreciated the reviewer’s point regarding the value of non-facial music blocks. However, although there are segments in music condition that have no faces presented, these cannot be used as a control condition to test whether the encoding model’s prediction accuracy in pSTC or DLPFC drops to chance when no facial emotion is present. This is because, in the absence of faces, no extracted emotion features are available to be used for the construction of encoding model (see Author response image 1 below). Thus, we chose to use a different control analysis for the present work. For children’s pSTC, we shuffled facial emotion feature in time to generate a null distribution, which was then used to test the statistical significance of the encoding models (see Methods/Encoding model fitting for details).

(1) Brooks, J. A. et al. Deep learning reveals what facial expressions mean to people in different cultures. iScience 27, 109175 (2024).

(2) Brooks, J. A. et al. Deep learning reveals what vocal bursts express in different cultures. Nat. Hum. Behav. 7, 240–250 (2023).

**Author response image 1. sa3fig1:** Time courses of Hume AI extracted facial expression features for the first block of music condition. Only top 5 facial expressions were shown here to due to space limitation.

(6) Generalizability is further limited by the fact that all participants were neurosurgical patients, potentially with neurological conditions such as epilepsy that may influence brain responses.

We appreciated the reviewer’s point. However, iEEG data can only be obtained from clinical populations (usually epilepsy patients) who have electrodes implantation. Given current knowledge about focal epilepsy and its potential effects on brain activity, researchers believe that epilepsy-affected brains can serve as a reasonable proxy for normal human brains when confounding influences are minimized through rigorous procedures [1]. In our study, we took several steps to ensure data quality: (1) all data segments containing epileptiform discharges were identified and removed at the very beginning of preprocessing, (2) patients were asked to participate the experiment several hours outside the window of seizures. Please see Method for data quality check description (Page 9/ Experimental procedures and iEEG data processing).

(1) Parvizi J, Kastner S. 2018. Promises and limitations of human intracranial electroencephalography. Nat Neurosci 21:474–483. doi:10.1038/s41593-018-0108-2

(7) Additionally, the high temporal resolution of intracranial EEG was not fully utilized, as data were down-sampled and averaged in 500-ms windows.

We agree that one of the major advantages of iEEG is its millisecond-level temporal resolution. In our case, the main reason for down-sampling was that the time series of facial emotion features extracted from the videos had a temporal resolution of 2 Hz, which were used for the modelling neural responses. In naturalistic contexts, facial emotion features do not change on a millisecond timescale, so a 500 ms window is sufficient to capture the relevant dynamics. Another advantage of iEEG is its tolerance to motion, which is excessive in young children (e.g., 5-year-olds). This makes our dataset uniquely valuable, suggesting robust representation in the pSTC but not in the DLPFC in young children. Moreover, since our method framework (Figure 1) does not rely on high temporal resolution method, so it can be transferred to non-invasive modalities such as fMRI, enabling future studies to test these developmental patterns in larger populations.

(8) Finally, the absence of behavioral measures or eye-tracking data makes it difficult to directly link neural activity to emotional understanding or determine which facial features participants afended to.

We appreciated this point. Part of our rationale is presented in our response to (5) for the absence of behavioral measures. Following the same rationale, identifying which facial features participants attended to is not necessary for testing our main hypotheses because our analyses examined responses to the overall emotional content of the faces. However, we agree and recommend future studies use eye-tracking and corresponding behavioral measures in studies of subjective emotional understanding.

**Reviewer #2 (Public review):**
Summary:In this paper, Fan et al. aim to characterize how neural representations of facial emotions evolve from childhood to adulthood. Using intracranial EEG recordings from participants aged 5 to 55, the authors assess the encoding of emotional content in high-level cortical regions. They report that while both the posterior superior temporal cortex (pSTC) and dorsolateral prefrontal cortex (DLPFC) are involved in representing facial emotions in older individuals, only the pSTC shows significant encoding in children. Moreover, the encoding of complex emotions in the pSTC appears to strengthen with age. These findings lead the authors to suggest that young children rely more on low-level sensory areas and propose a developmental shiZ from reliance on lower-level sensory areas in early childhood to increased top-down modulation by the prefrontal cortex as individuals mature.Strengths:(1) Rare and valuable dataset: The use of intracranial EEG recordings in a developmental sample is highly unusual and provides a unique opportunity to investigate neural dynamics with both high spatial and temporal resolution.(2) Developmentally relevant design: The broad age range and cross-sectional design are well-suited to explore age-related changes in neural representations.(3) Ecological validity: The use of naturalistic stimuli (movie clips) increases the ecological relevance of the findings.(4) Feature-based analysis: The authors employ AIbased tools to extract emotion-related features from naturalistic stimuli, which enables a data-driven approach to decoding neural representations of emotional content. This method allows for a more fine-grained analysis of emotion processing beyond traditional categorical labels.Weaknesses:(1) The emotional stimuli included facial expressions embedded in speech or music, making it difficult to isolate neural responses to facial emotion per se from those related to speech content or music-induced emotion.

We thank the reviewer for their raising this important point. We agree that in naturalistic settings, face often co-occur with speech, and that these sources of emotion can overlap. However, background music induced emotions have distinct temporal dynamics which are separable from facial emotion (See the Author response image 2 (A) and (B) below). In addition, face can convey a wide range of emotions (48 categories in Hume AI model), whereas music conveys far fewer (13 categories reported by a recent study [1]). Thus, when using facial emotion feature time series as regressors (with 48 emotion categories and rapid temporal dynamics), the model performance will reflect neural encoding of facial emotion in the music condition, rather than the slower and lower-dimensional emotion from music.

For the speech condition, we acknowledge that it is difficult to fully isolate neural responses to facial emotion from those to speech when the emotional content from faces and speech highly overlaps. However, in our study, (1) the time courses of emotion features from face and voice are still different (Author response image 2 (C) and (D)), (2) our main finding that DLPFC encodes facial expression information in postchildhood individuals but not in young children was found in both speech and music condition (Figure 2B and 2C). In music condition, neural responses to facial emotion are not affected by speech. Thus, we have included the DLPFC results from the music condition in the revised manuscript (Figure 2C), and we acknowledge that this issue should be carefully considered in future studies using videos with speech, as we have indicated in the future directions in the last paragraph of Discussion.

(1) Cowen, A. S., Fang, X., Sauter, D. & Keltner, D. What music makes us feel: At least 13 dimensions organize subjective experiences associated with music across different cultures. Proc Natl Acad Sci USA 117, 1924–1934 (2020).

**Author response image 2. sa3fig2:** Time courses of the amusement. (A) and (B) Amusement conveyed by face or music in a 30-s music block. Facial emotion features are extracted by Hume AI. For emotion from music, we approximated the amusement time course using a weighted combination of low-level acoustic features (RMS energy, spectral centroid, MFCCs), which capture intensity, brightness, and timbre cues linked to amusement. Notice that music continues when there are no faces presented. (C) and (D) Amusement conveyed by face or voice in a 30-s speech block. From 0 to 5 s, a girl is introducing her friend to a stranger. The camera focuses on the friend, who appears nervous, while the girl’s voice sounds cheerful. This mismatch explains why the shapes of the two time series differ at the beginning. Such situations occur frequently in naturalistic movies

(2) While the authors leveraged Hume AI to extract facial expression features from the video stimuli, they did not provide any validation of the tool's accuracy or reliability in the context of their dataset. It remains unclear how well the AI-derived emotion ratings align with human perception, particularly given the complexity and variability of naturalistic stimuli. Without such validation, it is difficult to assess the interpretability and robustness of the decoding results based on these features.

Hume AI models were trained and validated by human intensity ratings of large-scale, experimentally controlled emotional expression data [1-2]. The training process used both manual annotations from human raters and deep neural networks. Over 3000 human raters categorized facial expressions into emotion categories and rated on a 1-100 intensity scale. Thus, the outputs of Hume AI model reflect what typical facial expressions convey (based on how people actually interpret them), that is, the presented facial emotion. Our goal of the present study was to examine how facial emotions presented in the videos are encoded in the human brain at different developmental stages. We agree that the interpretation of facial emotions may be different in individual participants, resulting in different perceived emotion (i.e., the emotion that the observer subjectively interprets). Behavioral ratings are necessary to study the encoding of subjectively perceived emotion, which is a very interesting direction but beyond the scope of the present work. We have added text in the Discussion to explicitly note that our study focused on the encoding of presented emotion (second paragraph in Page 8).

(1) Brooks, J. A. et al. Deep learning reveals what facial expressions mean to people in different cultures. iScience 27, 109175 (2024).

(2) Brooks, J. A. et al. Deep learning reveals what vocal bursts express in different cultures. Nat. Hum. Behav. 7, 240–250 (2023).

(3) Only two children had relevant pSTC coverage, severely limiting the reliability and generalizability of results.

We appreciated this point and agreed with both reviewers who raised it as a significant concern. As described in response to reviewer 1 (comment 1), we have added data from another two children who have pSTC coverage. Group-level analysis using permutation test showed that children’s pSTC significantly encode facial emotion in naturalistic contexts (Figure 3B). Because iEEG data from young children are extremely rare, rapidly increasing the sample size within a few years is not feasible. However, we are confident in the reliability of our conclusion that children’s pSTC can encode facial emotion. First, the two new children’s responses (S33 and S49) from pSTC were highly consistent with our previous observations (see individual data in Figure 3B). Second, the averaged prediction accuracy in children’s pSTC (r_speech_=0.1565) was highly comparable to that in post-childhood group (r_speech_=0.1515).

(4) The rationale for focusing exclusively on high-frequency activity for decoding emotion representations is not provided, nor are results from other frequency bands explored.

We focused on high-frequency broadband (HFB) activity because it is widely considered to reflect the responses of local neuronal populations near the recording electrode, whereas low-frequency oscillations in the theta, alpha, and beta ranges are thought to serve as carrier frequencies for long-range communication across distributed networks[1-2]. Since our study aimed to examine the representation of facial emotion in localized cortical regions (DLPFC and pSTC), HFB activity provides the most direct measure of the relevant neural responses. We have added this rationale to the manuscript (Page 3).

(1) Parvizi, J. & Kastner, S. Promises and limitations of human intracranial electroencephalography. Nat. Neurosci. 21, 474–483 (2018).

(2) Buzsaki, G. Rhythms of the Brain. (Oxford University Press, Oxford, 200ti).

(5) The hypothesis of developmental emergence of top-down prefrontal modulation is not directly tested. No connectivity or co-activation analyses are reported, and the number of participants with simultaneous coverage of pSTC and DLPFC is not specified.

Directional connectivity analysis results were not shown because only one child has simultaneous coverage of pSTC and DLPFC. However, the Granger Causality results from post-childhood group (N=7) clearly showed that the influence in the alpha/beta band from DLPFC to pSTC (top-down) is gradually increased above the onset of face presentation (Author response image 3, below left, plotted in red). By comparison, the influence in the alpha/beta band from pSTC to DLPFC (bottom-up) is gradually decreased after the onset of face presentation (Author response image 3, below left, blue curve). The influence in alpha/beta band from DLPFC to pSTC was significantly increased at 750 and 1250 ms after the face presentation (face vs nonface, paired t-test, Bonferroni corrected P=0.005, 0.006), suggesting an enhanced top-down modulation in the post-childhood group during watching emotional faces. Interestingly, this top-down influence appears very different in the 8-year-old child at 1250 ms after the face presentation (Author response image 3, below left, black curve).

As we cannot draw direct conclusions from the single-subject sample presented here, the top-down hypothesis is introduced only as a possible explanation for our current results. We have removed potentially misleading statements, and we plan to test this hypothesis directly using MEG in the future.

**Author response image 3. sa3fig3:** Difference of Granger causality indices (face – nonface) in alpha/beta and gamma band for both directions. We identified a series of face onset in the movie that paticipant watched. Each trial was defined as -0.1 to 1.5 s relative to the onset. For the non-face control trials, we used houses, animals and scenes. Granger causality was calculated for 0-0.5 s, 0.5-1 s and 1-1.5 s time window. For the post-childhood group, GC indices were averaged across participants. Error bar is sem.

(6) The "post-childhood" group spans ages 13-55, conflating adolescence, young adulthood, and middle age. Developmental conclusions would benefit from finer age stratification.

We appreciate this insightful comment. Our current sample size does not allow such stratification. But we plan to address this important issue in future MEG studies with larger cohorts.

(7) The so-called "complex emotions" (e.g., embarrassment, pride, guilt, interest) used in the study often require contextual information, such as speech or narrative cues, for accurate interpretation, and are not typically discernible from facial expressions alone. As such, the observed age-related increase in neural encoding of these emotions may reflect not solely the maturation of facial emotion perception, but rather the development of integrative processing that combines facial, linguistic, and contextual cues. This raises the possibility that the reported effects are driven in part by language comprehension or broader social-cognitive integration, rather than by changes in facial expression processing per se.

We agree with this interpretation. Indeed, our results already show that speech influences the encoding of facial emotion in the DLPFC differently in the childhood and post-childhood groups (Figure 2D), suggesting that children’s ability to integrate multiple cues is still developing. Future studies are needed to systematically examine how linguistic cues and prior experiences contribute to the understanding of complex emotions from faces, which we have added to our future directions section (last paragraph in Discussion, Page 8-9).

**Recommendations for the authors:**

**Reviewer #1 (Recommendations for the authors):**
In the introduction: "These neuroimaging data imply that social and emotional experiences shape the prefrontal cortex's involvement in processing the emotional meaning of faces throughout development, probably through top-down modulation of early sensory areas." Aren't these supposed to be iEEG data instead of neuroimaging?

Corrected.

**Reviewer #2 (Recommendations for the authors):**
This manuscript would benefit from several improvements to strengthen the validity and interpretability of the findings:(1) Increase the sample size, especially for children with pSTC coverage.

We added data from another two children who have pSTC coverage. Please see our response to reviewer 2’s comment 3 and reviewer 1’s comment 1.

(2) Include directional connectivity analyses to test the proposed top-down modulation from DLPFC to pSTC.

Thanks for the suggestion. Please see our response to reviewer 2’s comment 5.

(3) Use controlled stimuli in an additional experiment to separate the effects of facial expression, speech, and music.

This is an excellent point. However, iEEG data collection from children is an exceptionally rare opportunity and typically requires many years, so we are unable to add a controlled-stimulus experiment to the current study. We plan to consider using controlled stimuli to study the processing of complex emotion using non-invasive method in the future. In addition, please see our response to reviewer 2’s comment 1 for a description of how neural responses to facial expression and music are separated in our study.